# Abnormalities of Hippocampal Subfield and Amygdalar Nuclei Volumes and Clinical Correlates in Behavioral Variant Frontotemporal Dementia with Obsessive–Compulsive Behavior—A Pilot Study

**DOI:** 10.3390/brainsci13111582

**Published:** 2023-11-11

**Authors:** Mu-N Liu, Li-Yu Hu, Chia-Fen Tsai, Chen-Jee Hong, Yuan-Hwa Chou, Chiung-Chih Chang, Kai-Chun Yang, Zi-Hong You, Chi Ieong Lau

**Affiliations:** 1Department of Psychiatry, Taipei Veterans General Hospital, Taipei 11217, Taiwan; liumuen@gmail.com (M.-N.L.); cjhong007@gmail.com (C.-J.H.);; 2School of Medicine, National Yang Ming Chiao Tung University, Taipei 11221, Taiwan; 3Center for Quality Management, Taipei Veterans General Hospital, Taipei 11217, Taiwan; 4Department of Neurology, Cognition and Aging Center, Kaohsiung Chang Gung Memorial Hospital, Kaohsiung 83301, Taiwan; 5College of Medicine, Chang Gung University, Taoyuan 33302, Taiwan; 6Department of Nephrology, Chiayi Branch, Taichung Veterans General Hospital, Chiayi 60090, Taiwan; 7Dementia Center, Department of Neurology, Shin Kong Wu Ho-Su Memorial Hospital, No.95, Wenchang Rd., Shilin Dist., Taipei 11101, Taiwan; 8Department of Neurology, University Hospital, Taipai, Macao SAR, China; 9Institute of Biophotonics, National Yang Ming Chiao Tung University, Taipei 11221, Taiwan; 10College of Medicine, Fu-Jen Catholic University, New Taipei City 24205, Taiwan; 11Applied Cognitive Neuroscience Group, Institute of Cognitive Neuroscience, 17 Queen Square, University College London, London WC1N 3AZ, UK

**Keywords:** behavioral variant frontotemporal dementia, obsessive–compulsive behavior, hippocampus, amygdala, MRI

## Abstract

(1) Background: The hippocampus (HP) and amygdala are essential structures in obsessive–compulsive behavior (OCB); however, the specific role of the HP in patients with behavioral variant frontotemporal dementia (bvFTD) and OCB remains unclear. (2) Objective: We investigated the alterations of hippocampal and amygdalar volumes in patients with bvFTD and OCB and assessed the correlations of clinical severity with hippocampal subfield and amygdalar nuclei volumes in bvFTD patients with OCB. (3) Materials and methods: Eight bvFTD patients with OCB were recruited and compared with eight age- and sex-matched healthy controls (HCs). Hippocampal subfield and amygdalar nuclei volumes were analyzed automatically using a 3T magnetic resonance image and FreeSurfer v7.1.1. All participants completed the Yale–Brown Obsessive–Compulsive Scale (Y-BOCS), Neuropsychiatric Inventory (NPI), and Frontal Behavioral Inventory (FBI). (4) Results: We observed remarkable reductions in bilateral total hippocampal volumes. Compared with the HCs, reductions in the left hippocampal subfield volume over the cornu ammonis (CA)1 body, CA2/3 body, CA4 body, granule cell layer, and molecular layer of the dentate gyrus (GC-ML-DG) body, molecular layer of the HP body, and hippocampal tail were more obvious in patients with bvFTD and OCB. Right subfield volumes over the CA1 body and molecular layer of the HP body were more significantly reduced in bvFTD patients with OCB than in those in HCs. We observed no significant difference in amygdalar nuclei volume between the groups. Among patients with bvFTD and OCB, Y-BOCS score was negatively correlated with left CA2/3 body volume (*τ*_b_ = −0.729, *p* < 0.001); total NPI score was negatively correlated with left GC-ML-DG body (*τ*_b_ = −0.648, *p* = 0.001) and total bilateral hippocampal volumes (left, *τ*_b_ = −0.629, *p* = 0.002; right, *τ*_b_ = −0.455, *p* = 0.023); and FBI score was negatively correlated with the left molecular layer of the HP body (*τ*_b_ = −0.668, *p* = 0.001), CA4 body (*τ*_b_ = −0.610, *p* = 0.002), and hippocampal tail volumes (*τ*_b_ = −0.552, *p* < 0.006). Mediation analysis confirmed these subfield volumes as direct biomarkers for clinical severity, independent of medial and lateral orbitofrontal volumes. (5) Conclusions: Alterations in hippocampal subfield volumes appear to be crucial in the pathophysiology of OCB development in patients with bvFTD.

## 1. Introduction

Behavioral variant frontotemporal dementia (bvFTD) is a progressive neurodegenerative constellation of signs and symptoms in which patients exhibit early personality changes and behavioral problems [1,2]. Despite being classified as one syndrome, bvFTD exhibits a wide variety of clinical features, making it a notoriously challenging diagnosis, particularly in its early stage [3]. Notably, behavioral symptoms precede cognitive deficits in bvFTD, differentiating it from Alzheimer’s disease (AD) [4,5,6,7]. Neuropathologically, protein aggregations and neuronal loss occur in the frontal and temporal lobes of patients with bvFTD [8]. Atrophy predominantly affects the frontal regions, including the prefrontal cortex and anterior cingulate, with extension to the anterior temporal lobes and amygdala [9]. Variability exists in atrophy patterns, including asymmetry and focal atrophy [10]. Furthermore, a variant known as the right temporal variant of frontotemporal dementia (rtv-FTD) primarily impacts the right temporal lobe, resulting in distinct symptoms like prosopagnosia and behavioral changes linked to right temporal dysfunction [11].

Among the variabilities in behavioral symptoms, the presence of obsessive–compulsive behavior (OCB), characterized by complex, repetitive, and time-consuming behaviors, is a common feature in patients with bvFTD [12]. One study indicated that 78% of pathologically proven frontotemporal dementia (FTD) involved behavioral disturbances such as OCB during a 1-year observation [13]. Another study revealed that OCB is present in nearly 38% of patients with FTD, while only approximately 10% of patients with AD exhibited behavioral disturbances [14]. Moreover, OCB poses significant challenges for caregivers, often leading to burnout [15,16]. Therefore, it is paramount to explore the pathophysiology of OCB in bvFTD patients.

While bvFTD is commonly associated with atrophy of the bilateral frontal and temporal lobes, there have been reports of asymmetric right-sided atrophy. This explanation is consistent with the mechanisms of the frontal lobes in behavioral control [17]. However, imaging studies have revealed that the patterns of anatomic changes associated with bvFTD extend beyond the frontal lobe to other brain regions, such as the hippocampus (HP) and amygdala [18,19]. Patients with bvFTD often exhibit varying patterns of atrophy corresponding to their specific behavioral problems [1,20]. Through analysis of the pathophysiology of OCB in addition to abnormalities in the frontal–subcortical circuit system [21,22], abnormal changes in the HP and amygdala have been emphasized in structural and functional imaging research [23,24]. Furthermore, medications used to manage OCB, such as serotonergic reuptake inhibitors, trigger neurogenesis and neuronal maturation in the HP [25], and the amygdala play a pivotal modulatory role in such activation [26]. 

Because abnormalities in behavioral problems may reflect variable patterns, structural changes in both the HP and amygdala have been implicated in OCB development. Thus, investigating these abnormal changes among bvFTD patients with OCB is crucial. This study aimed to fulfill two objectives. Firstly, we explored changes in hippocampal subfield and amygdalar nuclei volumes in patients with bvFTD and OCB compared to healthy controls (HCs). Secondly, we evaluated the correlation between hippocampal and amygdalar structural abnormalities and the severity of the behavioral symptoms implicating OCB. Furthermore, it is noteworthy that the orbitofrontal regions are among the initial sites of atrophy in bvFTD and play a crucial role in the development of clinical symptoms [27]. Our primary objective was to investigate whether hippocampal subfield volumes could directly reflect clinical severity. We sought to determine whether the influence of the hippocampal subfield on clinical severity is a direct relationship, rather than one mediated indirectly by the orbitofrontal cortex. We hypothesized that reductions in hippocampal subfield or amygdalar nuclei volumes could aid in identifying vulnerable regions in bvFTD patients with OCB. In addition, the severity of behavioral symptoms would be correlated with the selected substructural volumes in the HP and amygdala. This study aimed to provide crucial insights into the pathophysiology of OCB development in bvFTD patients.

## 2. Materials and Methods

### 2.1. Study Design

A comparative cross-sectional study.

### 2.2. Participants

Eight patients with suspected bvFTD who met international consensus criteria for diagnosis of bvFTD [10] and who presented to the Taipei Veterans General Hospital psychiatric department in Taiwan between 2020 and 2021 were included in the study. Patients with primary progressive aphasia and those unable to undergo magnetic resonance imaging (MRI) were excluded. All study participants had no premorbid history of obsessive–compulsive disorder (OCD), and none of them had been diagnosed as having OCD on the basis of DSM-V criteria. 

OCB was defined as behavioral problems that could be identified in accordance with the Yale–Brown Obsessive–Compulsive Scale (Y-BOCS) symptom checklist [28]. The Y-BOCS includes over 50 types of obsessions or compulsions divided into 15 categories derived from behavioral patterns observed in patients with OCD. Advertisements were used to recruit eight age-matched and sex-matched HCs without any neurological or psychiatric conditions from the community. In the HC group, none exhibited any signs of dementia. All individuals had a Clinical Dementia Rating (CDR) [29] score of zero, and their Mini–Mental Status Examination (MMSE) [30] scores exceeded 25. In addition, none of the individuals in the HC group had any family history of neurological or psychiatric diseases, and they were not taking any medications that would affect brain volume. The study was approved by the Institutional Review Board of Taipei Veterans General Hospital and was conducted in compliance with the principles outlined in the Declaration of Helsinki. Written informed consent was obtained from all the participants.

### 2.3. Neuropsychiatric Assessment

Both groups completed neuropsychological tests to evaluate various cognitive functions. The CDR (with a cutoff of 1 indicating mild impairment and higher scores indicating more severe impairment) [29] and the MMSE (with a maximum score of 30, and scores below 24 indicating cognitive impairment) [30] were employed to test the cognitive status of participants. The MMSE was standardized for their specific language and population [31]. The Y-BOCS was used to assess the severity of OCB, using a continuous measure rather than specific cutoff points [28]. The Neuropsychiatric Inventory (NPI) was used to determine noncognitive behavioral and psychiatric disturbances [32], with a higher score indicating more severe behavioral and psychiatric symptoms. The 24-item Frontal Behavioral Inventory (FBI) was used to evaluate personality changes and behavioral problems related to bvFTD, with a higher score indicating more severe behavioral symptoms [33,34]. Responses to questionnaires were obtained from the caregivers of the patients. All assessments were completed on the same date on which the participants underwent an MRI scan.

### 2.4. MRI Acquisition

Structural MRI data were gathered using a 3T MR750 scanner (GE Medical Systems, Milwaukee, WI, USA). T1-weighted anatomical MRI data were collected using the following parameters: brain volume (“BRAVO”) sequence; repetition time = 12.2 ms; echo time = 5.2 ms; inversion time = 450 ms; flip angle = 12°; matrix size = 256 × 256; field of view = 256 mm; 172 axial slices with 1 mm slice thickness; and resolution = 1 × 1 × 1 mm^3^.

### 2.5. Analysis of Hippocampal Subfields and Amygdalar Nuclei Volumes

Automatic volumetric quantification of hippocampal and amygdalar structures was performed using FreeSurfer v7.1.1, with the hippocampus/amygdala module (Massachusetts General Hospital, Boston, MA, USA; http://surfer.nmr.mgh.harvard.edu, accessed on 27 July 2020) used on the T1-weighted image. The HP was segmented into the following 19 subfields: the subiculum, presubiculum, parasubiculum, molecular layer, cornu ammonis (CA)1, CA2/3, CA4, hippocampal tail, hippocampal fissure, fimbria, HP–amygdala transition area, and granule cell layer and molecular layer of the dentate gyrus (GC-ML-DG). The subiculum, presubiculum, GC-ML-DG, molecular layer, CA1, CA2/3, and CA4 were further segmented into head and body portions (Figure 1). 

In total, nine nuclei were segmented for the amygdala, namely the lateral, basal, central, medial, cortical, accessory basal, cortico-amygdaloid transition area, anterior amygdaloid area, and paralaminar nuclei (Figure 1). The details of the automated segmentation procedure for quantifying hippocampal and amygdalar volumes have been described in other studies [35,36].

### 2.6. Mediation Analysis

To understand whether hippocampal subfield volumes can directly serve as clinical markers or if their relationship with clinical severity is influenced by indirect pathways, such as lateral or medial orbital frontal volumes, we conducted a comprehensive mediation analysis [37]. In this analysis, hippocampal subfield volumes were designated as predictors, while medial and lateral orbitofrontal volumes were identified as mediators, and clinical test scores were established as the outcomes. To ensure the robustness of our findings, we employed bootstrapping tests with 1000 resamples and applied bias-corrected confidence intervals to construct and validate the mediation model. In our model, two distinct pathways were examined: the total direct pathway represented by hippocampal subfield volumes → clinical severity, and the indirect pathway encompassing hippocampal subfield volumes → orbitofrontal volumes → clinical severity. The clinical outcomes evaluated were the Y-BOCS, total NPI score, and FBI score. We applied a significance level of *p* < 0.05 to determine the statistical significance of our analyses.

### 2.7. Statistical Analysis

Statistical analyses were performed using SPSS v.19 (IBM, Armonk, NY, USA). We conducted normality tests (Kolmogorov–Smirnoff and Shapiro–Wilk tests) to assess distribution (Appendix A). Continuous variables, such as demographic data, were analyzed using an independent *t*-test or Mann–Whitney U test if the data were not in a normal distribution. Effect size was calculated by Cohen’s d or Mann–Whitney effect size (r). Hippocampal subfield and amygdalar nuclei volumes were analyzed using a general linear model. Individual volume was regarded as a dependent variable, diagnosis as an independent variable, and age, sex, and total intracranial volume as covariates. We applied Bonferroni corrections to adjust all the results and used *p* < 0.05/*n* (*n* is the comparison time) to indicate significance. Kendall rank correlation was adopted to explore the association between individual volume and clinical severity (i.e., Y-BOCS, total NPI, and FBI scores). 

## 3. Results

We enrolled eight patients with bvFTD and OCB (mean age = 66.0 ± 5.7 years) and eight age-matched and sex-matched HCs (mean age = 64.5 ± 5.3 years). Age and sex were not significantly different between the groups (Table 1). Among patients with bvFTD, the CDR score was 1 (indicating mild severity). The mean Y-BOCS score was 24.1 ± 1.6, with three patients displaying holding and collecting behavior, three patients demonstrating repetitive checking behavior (related to time, locks, and clothing), and two patients engaging in cleaning and washing behaviors. The total NPI score was 21.4 ± 5, and the FBI score was 14.9 ± 2.9. 

Cohen’s d equation is shown below.
$$Cohen’s\;d=\frac{\mathrm{group}\;A\;\mathrm{means}\;-\;\mathrm{group}\;B\;\mathrm{means}}{pooled\;standard\;deviation}$$

Effect size analysis of Mann–Whitney U was calculated using the *r* equation as follows:$$r=\frac{Z}{\surd N}$$

Regarding hippocampal subfield volumes, compared with HCs, we observed remarkable reductions in bilateral hippocampal volumes (left, F(1, 11) = 24.724, *p* < 0.001; right, F(1, 11) = 21.066, *p* = 0.001); in the left subfield volumes over the CA1 body (F(1, 11) = 18.943, *p* = 0.001), molecular layer of the HP body (F(1, 11) = 38.939, *p* < 0.001), CA2/3 body (F(1, 11) = 40.097, *p* < 0.001), GC-ML-DG-body (F(1, 11) = 21.259, *p* = 0.001), CA4 body (F(1, 11) = 23.133, *p* = 0.001), and hippocampal tail (F(1, 11) = 30.928, *p* < 0.001); and in the right subfield volumes over the CA1 body (F(1, 11) = 21.448, *p* = 0.001) and molecular layer of the HP body (F(1, 11) = 20.342, *p* = 0.001) in patients with bvFTD and OCB (Table 2). We observed no statistically significant differences in the volumes of bilateral amygdalar nuclei between the patients with bvFTD and OCB and the HCs (Table 3).

Among patients with bvFTD and OCB, Y-BOCS score was negatively correlated with left CA2/3 body volume (*τ*_b_ = −0.729, *p* < 0.001; Figure 2A); total NPI score was negatively correlated with left GC-ML-DG body (*τ*_b_ = −0.648, *p* = 0.001) and bilateral total hippocampal volumes (left, *τ*_b_ = −0.629, *p* = 0.002; right, *τ*_b_ = −0.455, *p* = 0.023; Figure 2B); and FBI score was negatively correlated with left molecular layer of the HP body (*τ*_b_ = −0.668, *p* = 0.001), CA4 body (*τ*_b_ = −0.610, *p* = 0.002), and hippocampal tail volumes (*τ*_b_ = −0.552, *p* < 0.006; Figure 2C). Mediation analysis was performed to explore whether these subfield volumes could serve as direct biomarkers for clinical severity. All predefined clinical outcomes were directly influenced by the hippocampal subfield volumes without mediation by bilateral medial and lateral orbitofrontal volumes (Appendix A).

## 4. Discussion 

This is the first study exploring changes in hippocampal subfield and amygdalar nuclei volumes of bvFTD patients with OCB. The main findings are as follows: (1) reduced volumes over the bilateral HP and several specific hippocampal subfields, especially in the body regions (left subfield: CA1 body, molecular layer of the HP body, CA2/3 body, GC-ML-DG body, CA4 body, and hippocampal tail; right subfield: CA1 body and molecular layer of the HP body); (2) no significant volume changes in total amygdala or any amygdalar nuclei between patients with bvFTD and OCB and HCs; and (3) a statistically significant negative correlation of hippocampal subfield volumes with several domains of clinical symptom severity, including the severity of OCB. The study results were as follows: (i) left CA2/3 body volume negatively correlated with OCB severity (Y-BOCS score); (ii) left molecular layer of the HP body, CA4 body, and hippocampal tail volumes negatively correlated with the severity of behavioral symptoms (FBI score); and (iii) left GC-ML-DG body and bilateral total hippocampal volumes negatively correlated with the severity of neuropsychiatric symptoms (total NPI score). Studies have established a close correlation between clinical symptoms and both hippocampal and orbitofrontal volumes [27]. Our study extends these findings by suggesting that the hippocampus affects a broader range of clinical severity which is not mediated by the orbitofrontal cortex. Thus, we concluded that volume reductions in hippocampal subfields, especially the left hippocampal body, occur in patients with bvFTD and OCB. Such structural changes were also correlated with the severity of neuropsychiatric behavioral disturbances.

This study revealed significant volume reductions in bilateral hippocampal subfields among patients with bvFTD and OCB. Numerous studies have reported that the HP is an essential brain structure involved in the development of OCB. Szezsko et al. [38] and Hong et al. [39] have also studied patients with bvFTD and OCB, revealing that left and right hippocampal volumes were reduced compared with those of HCs. Atmaca and colleagues [24] indicated that patients with refractory OCD had smaller bilateral hippocampal volumes and revealed a correlation between clinical severity and the left HP. Rosso and colleagues [12] reported a similar finding, determining that left temporal lobe atrophy plays a pivotal role in the pathophysiology of compulsive behavior development in patients with FTD. A SPECT study of FTD revealed that left temporal hypoperfusion was associated with compulsions [40]. In bvFTD, a pathological hallmark is the abnormal accumulation of proteins, particularly TDP-43, which disrupts crucial cellular functions, notably RNA processing, ultimately leading to neurodegeneration [1]. This process also impacts hippocampal integrity [41]. Furthermore, individuals with bvFTD exhibiting OCB often exhibit a distinct reduction in hippocampal volume [42]. In cases of carriers with MAPT mutations, significant atrophy is frequently observed in the medial temporal area, specifically affecting the anterior and central portions of the hippocampus. These regions are integral components of the limbic system and their impairment is associated with challenges in emotional regulation [43]. Another study showed that those with MAPT mutations had marked volumetric differences in the hippocampus proper, which include the CA subfields. Conversely, those with C9orf72 expansions exhibit the most substantial atrophy in the dentate gyrus and CA1/4. Meanwhile, individuals with GRN mutations demonstrate the greatest impact on the subiculum and presubiculum [44]. In the case of svPPA, both the CA1 and subiculum regions displayed substantial reductions in volume compared to controls [45]. 

Given the specific structural connections between hippocampal projections and the frontal cortex in both the HP and prefrontal region, the role of the HP in OCB could be more fully understood from a neuroanatomic perspective [46]. Serotonin reuptake inhibitors are a common treatment choice for compulsive behaviors in patients with bvFTD and OCB, as they have been shown to bind to many serotonin receptors [25]. In summary, we observed significant volumetric reductions in bilateral hippocampal subfields, especially in the left subfields of the HP body, in patients with bvFTD and OCB. Our findings indicate that the HP could play an essential role in the pathophysiology of OCB development in patients with bvFTD. 

In addition, we noted greater volume reductions in hippocampal subfields over the CA1, CA2/3, CA4, GC-ML-DG, molecular layer of the HP body, and hippocampal tail in the left hemisphere and in those over the CA1 and molecular layer of the HP body in the right hemisphere in patients with bvFTD and OCB. We obtained a clearer view of the HP as a heterogeneously functional brain structure along the body’s anterior–posterior axis (head, body, and tail). Similar to the dorsal HP in rats, the posterior HP is associated with cognitive functions such as managing spatial information, learning, and cognitive flexibility, which support the demonstration of flexible goal-directed behaviors and habit learning [47]. Hence, atrophy of the hippocampal body and tail subfields may be related to impaired functions in the balance between goal-directed behavior and habit learning, which represent a crucial neurocognitive feature of OCB [48,49]. The anterior HP exhibits volumetric reductions before the posterior HP as part of the normal aging process [50]. Therefore, in the older adult population, the functional connectivity of the anterior HP may be impaired, with the posterior hippocampal functional connectivity compensating through increased connectivity to neocortical regions [51]. The posterior HPs of older adults exhibited more connectivity to several brain regions, including the cuneus, precuneus, and cingulum [51]. These areas are involved in the development of compulsivity [52]. Hence, volumetric reductions over the subfields of the hippocampal body and tail in patients with bvFTD may impair the efficiency of the connectivity between the HP and these specific brain areas that are responsible for compulsive behaviors.

The absence of group differences in both total amygdalar volume and amygdalar nuclei volumes in this study contradicts the findings of other studies. Several studies have reported amygdalar volume reductions in patients with bvFTD [53], as well as in those with OCD [24,38], and in FTD patients with OCB [42]. However, other studies have revealed no significant reduction in amygdalar volume in patients with FTD [9] or those with OCD [54]. These inconsistent findings may be the result of several factors, such as different pathological behavioral symptom profiles [55,56]. In addition, the small sample size in our study could have led to an underestimation of amygdalar volumetric reduction. Additional studies with larger sample sizes are required. Researchers could recruit more homogeneous participants to clarify this inconsistency and to explore the probable mechanisms underlying amygdalar nuclei volumetric changes associated with OCB among patients with bvFTD. 

There appears to be a left-sided predominance in the structural changes in our study, encompassing the entire hippocampus and specific subfields: CA1-body, CA2/3-body, CA4-body, GC-ML-DG-body, and Molecular_layer_HP-body, as well as the hippocampal tail. The left temporal lobe is implicated in the compulsive behaviors of patients with FTD. A SPECT study of FTD revealed that left temporal hypoperfusion was associated with compulsions and mental rigidity [40]. In another study, compulsive severity, as assessed using total Y-BOCS score, was correlated with grey matter loss in the left temporal lobe in patients with OCB and bvFTD [57]. In addition, the subfield volume of the CA2/3 was correlated inversely with the severity of OCB symptoms in patients with several neuropsychiatric disorders [58]. Connectivity within the CA3 subfield region is more complicated and richer than in other hippocampal areas. The intricate axon network of CA3 pyramidal cells generates considerable ramifications, inducing interactions between excitatory and inhibitory neurons. This circuit is involved in spatial representations and memories [59]. Deficits of these cognitive processes are observed in bvFTD [60,61]. Notably, similar mechanisms may play a role in the pathophysiology of OCB [62].

We observed a negative correlation between FBI scores and the volumes of the left molecular layer of the HP body, CA4 body, and hippocampal tail. The FBI was designed to evaluate personality changes and behavioral problems in patients with bvFTD, and FBI scores are highly correlated with frontal lobe function [63]. Both clinical and animal studies have revealed that hippocampal subfields, including the molecular layer of the HP body, CA4, and hippocampal tail, interact. Because the HP is connected to the frontal lobe [64], the FBI can be used to determine whether a central volumetric reduction in this area in patients with bvFTD is responsible for behavioral disturbance [33]. Therefore, hippocampal subfield volume may be negatively correlated to the severity of clinical behavioral symptoms through its complicated interaction with the frontal region. In the current study, we tested whether the relationship between the volumes of these three subfields and FBI severity is mediated by the orbitofrontal cortex. We found that the volumes of these subfields had a direct effect on FBI severity without mediation by orbitofrontal cortex volume, which is consistent with a prior report suggesting that the hippocampus directly affects frontal behavior change [65]. On the other hand, the frontal lobe is a large region containing several parts other than the orbitofrontal cortex (e.g., prefrontal cortex) and has extensive connections with various other brain regions (e.g., hypothalamus), which may be responsible for frontal behavior changes [66,67]. The hippocampus may interact with these regions to influence frontal behavior, but not the orbitofrontal cortex [68]. Future studies need to clarify the underlying mechanism of the relationship between hippocampal subfield volume and FBI severity. 

Finally, we observed a negative correlation between total NPI score and the subfield volumes of the left dentate gyrus (DG) and total bilateral HP. The DG is involved in neurogenesis, a process where new neurons are continuously generated throughout adult life. As a result, the DG may be more vulnerable to stress-related toxic damage and could be associated with the development of neuropsychiatric symptoms [69]. Animal studies have shown that inhibiting the pyramidal neurons of the DG is required to suppress neuropsychiatric symptoms [70]. A systematic review and meta-analysis also indicated that the level of dysfunction or volume deficit in the DG is associated with the severity of neuropsychological symptoms [71]. Given the anatomical and functional complexity of the HP, an analysis of hippocampal subfield volumes offers insight into the specific hippocampal changes associated with the severity of the neuropsychiatric symptoms in bvFTD, particularly OCB. In summary, the HP is a heterogeneous structure that consists of several anatomically and functionally distinct subfields, which may contribute to distinctive clinical presentations, including that of OCB in patients with bvFTD.

This study has several limitations. First, because of the cross-sectional study design, clarifying causal relationships was difficult. Volumetric changes in the HP could either be an essential factor in the development of OCB or be a consequence of structural changes resulting from OCB in bvFTD. A prospective study could investigate this further. Second, the small sample size of both groups may have led to bias. Therefore, the results of this study should be considered preliminary. A large sample size would help in assessing subtle changes in subfield volume in the HP and amygdala. Third, the clinical diagnoses of our patients with bvFTD were not pathologically or genetically verified. Thus, patients diagnosed as having bvFTD might have had a frontal variant of AD or a mixed pathology with other etiologies [72]. Fourth, we did not control for the medication administered to the patients with bvFTD and OCB and thus could not account for the variety of medications and the possibility of poor drug compliance. Medication types, dosages, and administration durations may affect HP and amygdala volumes [73,74,75]. Thus, the effects of medication on the study results cannot be excluded. Fifth, we only enrolled patients with bvFTD and OCB. Therefore, distinguishing which hippocampal subfield changes are specific to bvFTD and which are specific to OCB is difficult. To clarify this, studies could compare patients with bvFTD with and without OCB. Finally, utilizing different techniques to calculate hippocampal subfield volumes may generate different results [76]. The study used the automatic method of FreeSurfer v7.1.1, which may yield discrepant results compared to studies using alternative techniques. However, FreeSurfer morphometric procedures have strong test–retest reliability across various scanners and field strengths [77]. Future studies could explore various advanced automatic segmentation approaches to ensure consistent results and determine the most feasible method for measuring hippocampal subfield volume in patients with bvFTD and OCB. 

## 5. Conclusions

We observed hippocampal subfield volumetric reductions in patients with bvFTD and OCB. We noted statistically significant inverse correlations between hippocampal subfield volumes and scale scores associated with behavioral symptoms, especially OCB. This verified the pivotal role of the HP in the pathogenesis and phenomenology of bvFTD with OCB. In summary, the observed changes in hippocampal subfield volumes appear to be significant in the pathophysiology of OCB development in patients with bvFTD. It would be valuable for future studies to further investigate whether these changes could serve as a potential imaging biomarker for diagnosing bvFTD with OCB.

## Figures and Tables

**Figure 1 brainsci-13-01582-f001:**
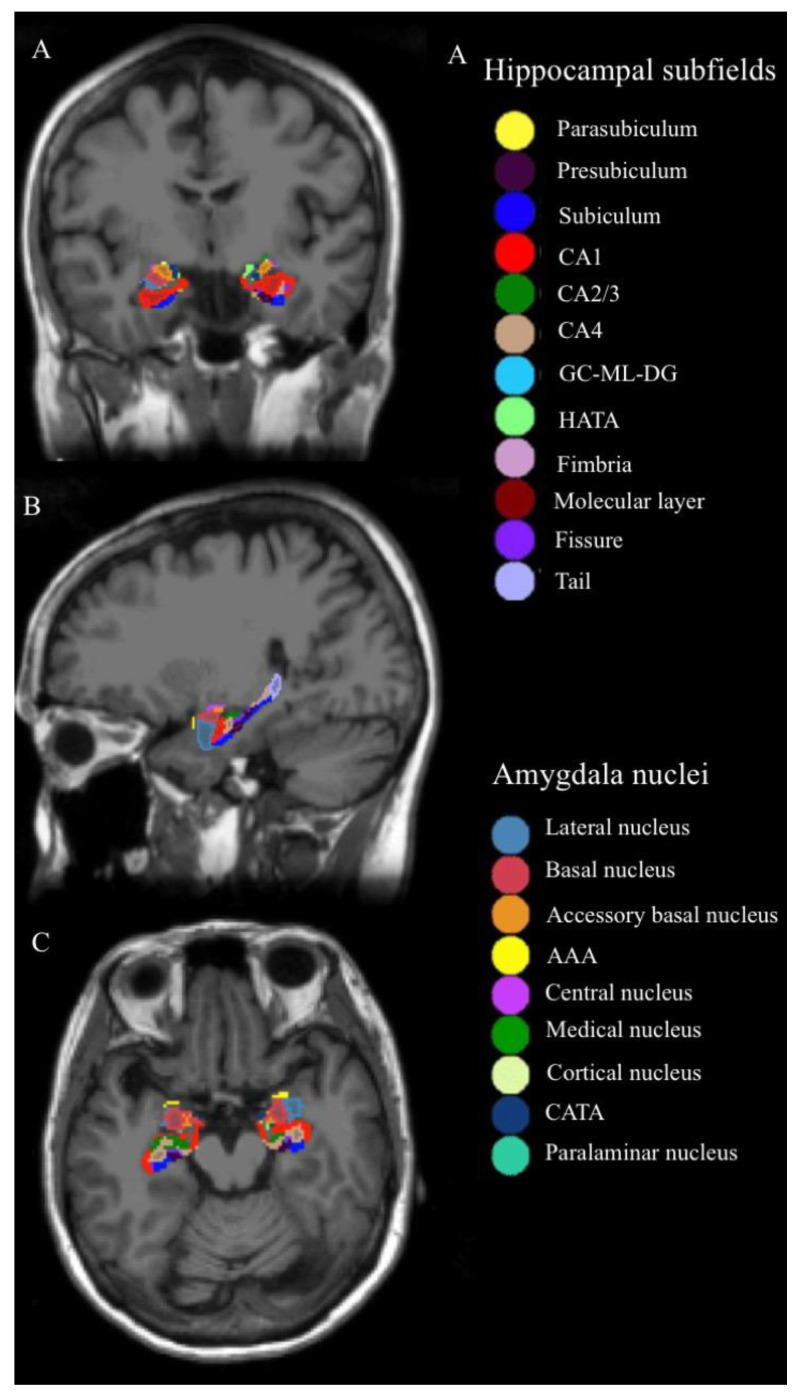
Hippocampal subfields and amygdala nuclei of one of the subjects: the column of (**A**–**C**) represents coronal, sagittal, and axial image views, respectively. Abbreviations: CA, cornu ammonis; GC-ML-DG, granule cell layer and molecular of the dentate gyrus; HATA, hippocampus–amygdala transition area; AAA, anterior amygdaloid area; CATA, cortico-amygdaloid transition area.

**Figure 2 brainsci-13-01582-f002:**
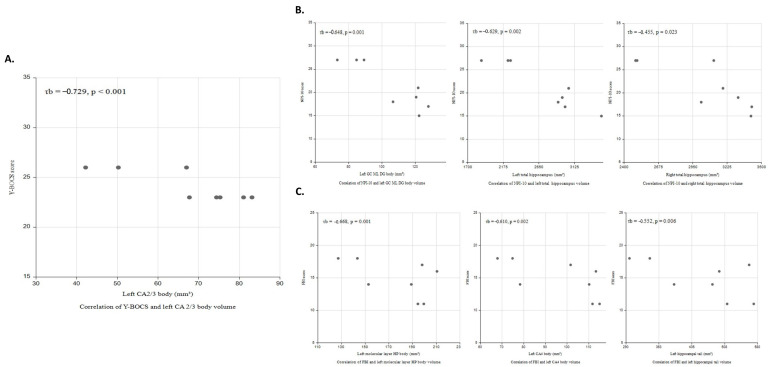
Correlations of clinical severity with hippocampal subfield volume in bvFTD patients with OCB. (**A**) Scatter plots for the relationship between Y-BOCS and left CA2/3 body volume (*τ*_b_ = −0.729, *p* < 0.001); (**B**) total NPI-10 score was negatively correlated with left GC ML DG body (*τ*_b_ = −0.648, *p* = 0.001) and bilateral total hippocampal volumes (left (*τ*_b_ = −0.629, *p* = 0.002), right (*τ*_b_ = −0.455, *p* = 0.023)); and (**C**) FBI score was negatively correlated with left molecular layer body (*τ*_b_ = −0.668, *p* = 0.001), CA4 body (*τ*_b_ = −0.610, *p* = 0.002), and tail volumes (*τ*_b_ = −0.552, *p* < 0.006).

**Table 1 brainsci-13-01582-t001:** Demographic and clinical variables for bvFTD patients with obsessive–compulsive behaviors and controls.

		bvFTD with OCB (*n* = 8)	Controls (*n* = 8)	*p*	Effect Size
Age, year		66.0 ± 5.7	64.5 ± 5.3	0.598	0.27
Sex (Female/Male)		6/2	6/2	0.715	
Education, year		6.0 ± 1.1	6.1 ± 1.0	0.798	0.08
Handedness (right)		8	8		
MMSE		20.1 ± 6.9	30 ± 0	0.002	0.81
CDR		1 ± 0	0		
Y-BOCS			0		
	Total	24.1 ± 1.6	0		
	Obsession	8.8 ± 1.0	0		
	Compulsion	15.4 ± 0.5	0		
NPI-10					
	Total	21.4 ± 5.0	0		
	Severity	9.5 ± 0.5	0		
	Frequency	11.4 ± 1.6	0		
FBI					
	FBI-total	14.9 ± 2.9	0		
	Negative	3.1 ± 0.8	0		
	Disinhibition	11.8 ± 2.2	0		
TIV		1,495,629 ± 254,069	1,418,600 ± 199,993	0.511	0.34

Values denote mean ± standard deviation or *n* (%). Median and interquartile range (Q1–Q3) of education in bvFTD with OCD: 6.0 (5–6.8); and controls: 6.0 (5.3–6.8); MMSE: bvFTD with OCD: 22 (12–25); and controls: 30 (30–30). Abbreviations: OCB, obsessive–compulsive behaviors; MMSE, Mini–Mental Status Examination; CDR, Clinical Dementia Rating; NPI, Neuropsychiatric Inventory; FBI, Frontal Behavioral Inventory; Y-BOCS, Yale–Brown Obsessive–Compulsive Scale; TIV, total intracranial volume. All volumes are in cubic millimeters (mm^3^). Chi-square for sex, *t*-test for age and TIV, Mann–Whitney test for education. Cohen’s d was used to estimate sample sizes for age and TIV, while Mann–Whitney effect size (r) was used for education and MMSE.

**Table 2 brainsci-13-01582-t002:** Gray matter volume in subfields of hippocampus.

	Subfields (mm^3^)	bvFTD with OCB (*n* = 8)	Controls (*n* = 8)	GLM, Age, Sex, TIV
Left	Parasubiculum	52 ± 20	60 ± 10	0.142
	Presubiculum-head	106 ± 32	139 ± 17	0.011
	Presubiculum-body	144 ± 38	156 ± 16	0.463
	Subiculum-head	154 ± 35	191 ± 25	0.008
	Subiculum-body	203 ± 47	242 ± 22	0.026
	CA1-head	411 ± 80	500 ± 68	0.005
	CA1-body	98 ± 26	140 ± 25	0.001 *
	CA2/3-head	87 ± 26	112 ± 20	0.017
	CA2/3-body	68 ± 15	98 ± 16	<0.001 *
	CA4-head	97 ± 25	122 ± 15	0.009
	CA4-body	97 ± 20	123 ± 13	0.001 *
	GC-ML-DG-head	114 ± 28	147 ± 21	0.006
	GC-ML-DG-body	106 ± 21	136 ± 13	0.001 *
	Molecular_layer_HP-head	258 ± 57	322 ± 37	0.004
	Molecular_layer_HP-body	177 ± 31	232 ± 23	<0.001 *
	HATA	44 ± 10	52 ± 8	0.013
	Fimbria	51 ± 21	71 ± 14	0.027
	Hippocampal tail	457 ± 101	593 ± 79	<0.001 *
	Hippocampal fissure	139 ± 22	152 ± 32	0.053
	Whole hippocampus	2724 ± 535	3437 ± 336	<0.001 *
Right	Parasubiculum	56 ± 14	55 ± 8	0.773
	Presubiculum-head	116 ± 16	128 ± 16	0.026
	Presubiculum-body	140 ± 19	146 ± 14	0.420
	Subiculum-head	167 ± 25	186 ± 27	0.007
	Subiculum-body	210 ± 24	237 ± 35	0.015
	CA1-head	468 ± 66	526 ± 75	0.004
	CA1-body	110 ± 22	139 ± 15	0.001 *
	CA2/3-head	107 ± 16	122 ± 19	0.024
	CA2/3-body	83 ± 17	100 ± 13	0.003
	CA4-head	119 ± 17	129 ± 18	0.055
	CA4-body	110 ± 19	127 ± 18	0.004
	GC-ML-DG-head	140 ± 22	154 ± 23	0.032
	GC-ML-DG-body	120 ± 222	137 ± 17	0.004
	Molecular_layer_HP-head	292 ± 40	330 ± 42	0.003
	Molecular_layer_HP-body	195 ± 27	231 ± 22	0.001 *
	HATA	47 ± 9	56 ± 8	0.019
	Fimbria	65 ± 12	66 ± 21	0.261
	Hippocampal tail	514 ± 75	599 ± 99	0.004
	Hippocampal-fissure	1622 ± 30	174 ± 41	0.156
	Whole hippocampus	3060 ± 374	3469 ± 398	0.001 *

Data in the second and third columns of the table represent the mean volume ± standard deviation. Unit: mm^3^. Abbreviations: GLM, generalized linear model; TIV, total intracranial volume; OCB, obsessive–compulsive behaviors; CA, cornu ammonis; HP, hippocampal; GC-ML-DG, granule cell molecular layer of dentate gyrus; and HATA, hippocampus–amygdala transition area. Results were corrected by Bonferroni correction with * *p* < 0.05/40 = 0.00125.

**Table 3 brainsci-13-01582-t003:** Gray matter volume in nuclei of amygdala.

	Amygdala Nuclei (mm^3^)	bvFTD with OCB (*n* = 8)	Controls (*n* = 8)	GLM, Age, Sex, TIV
Left	Lateral nucleus	565 (130)	654 (113)	0.037
	Basal nucleus	329 (81)	403 (62)	0.022
	Accessory basal nucleus	183 (46)	235 (33)	0.026
	Anterior amygdaloid area	41 (9)	52 (9)	0.011
	Central nucleus	30 (9)	39 (7)	0.055
	Medial nucleus	15 (4)	19 (5)	0.111
	Cortical nucleus	17 (5)	24 (4)	0.021
	CATA	128 (27)	155 (26)	0.042
	Paralaminar nucleus	39 (10)	47 (8)	0.033
	Whole amygdala	1346 (303)	1627 (246)	0.022
Right	Lateral nucleus	589 (95)	650 (81)	0.073
	Basal nucleus	378 (59)	419 (59)	0.033
	Accessory basal nucleus	216 (39)	254 (38)	0.023
	Anterior amygdaloid area	49 (8)	56 (9)	0.086
	Central nucleus	35 (8)	43 (8)	0.058
	Medial nucleus	20 (8)	22 (7)	0.347
	Cortical nucleus	21 (4)	26 (5)	0.032
	CATA	143 (27)	164 (24)	0.050
	Paralaminar nucleus	43 (7)	47 (7)	0.094
	Whole amygdala	1494 (235)	1679 (226)	0.039

Data in the second and third columns of the table represent the mean volume ± standard deviation. Unit: mm^3^. The results were corrected by Bonferroni correction with *p* < 0.05/20 = 0.0025. Abbreviations: GLM, generalized linear model; TIV, total intracranial volume; OCB, obsessive–compulsive behaviors; CATA, cortico-amygdaloid transition area.

## Data Availability

The original contributions presented in the study are included in the article; further inquiries can be directed to the corresponding author.

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
