# Peer review of "Abnormalities of Hippocampal Subfield and Amygdalar Nuclei Volumes and Clinical Correlates in Behavioral Variant Frontotemporal Dementia with Obsessive–Compulsive Behavior—A Pilot Study"

_brainsci, 2023, doi:10.3390/brainsci13111582_

Round 1
Reviewer 1 Report
Comments and Suggestions for Authors
The paper by Mu-N Liu et al. is an interesting study evaluating the correlation between volumetric changes in the hippocampus and amygdala nuclei and the presence and severity of OCB in patients affected by bvFTD, compared to healthy controls. This is promising work, but the extremely small sample size and the presence of other limitations (correctly reported by the Authors) such as the absence of bvFTD patients without OCB, makes it difficult to establish a causal link between the measures investigated. I have some suggestions for the Authors:
- Lines 69-70: I suggest to rephrase this sentence, in particular by describing in more detail the neuropathological changes and the pattern of atrophy in patients with bvFTD; in addition, the right lobe variant of FTD has been introduced in recent years, so I advise the Authors to make a brief description of this by citing the relevant literature on the subject;
- I suggest including in the title in the methods, e.g. 'a pilot study' or a similar add considering the small sample size;
- Methods: there are various aspects to be integrated: what are the cut-off scores to indicate that patients with bvFTD have OCB and HCs do not?
-Were patients with bvFTD assessed from a genetic point of view, as some mutations are known to be more associated with OCB than others? If not, it should be added to the limitations of the study;
- Regarding healthy controls: MMSE score > 25 does not necessarily imply normality, as some scales indicate between 27 and 24 a condition of MCI. Were the HC really healthy or could they have mild cognitive impairment, even if not full-blown dementia? If so, what might be the implications? Furthermore, did HCs have a family history of dementia? Did they have a history of neurological diseases? Were they taking drugs that could have an effect on the volume of the regions under investigation?
- Are MMSE scores also available for patients with bvFTD?
- I suggest adding, if correct, that the study was conducted in accordance with the Declaration of Helsinki by citing the relevant bibliography;
- I suggest describing in more detail the tests used (e.g. MMSE, CDR, NPI, FBI) and the cut-offs used are by citing the relevant literature.
- With regard to the results, it would be useful to assess the aspect of lateralisation of volumetric changes (i.e. right vs. left) in more detail. Did the Authors expect a prevalence of alterations on one side rather than the other? Why?
- The Authors rightly described that the lack of bvFTD patients without OCB makes it difficult to attribute a causal link between volumetric changes and severity of symptoms. However, in this way, it is possible that the greater severity of symptoms is simply associated with a greater severity of the neurodegenerative process in the brain, rather than with damage to specific areas examined. Could the Authors identify an area that, hypothetically, should be poorly impaired (especially in the early stages of the neurodegenerative process, as the low CDR scores seem to indicate) and use it as a comparator to make the causal link between hippocampal alterations and neuropsychological measures more robust?
- In the Discussion, I suggest providing some details on the neuropathological and molecular mechanisms of FTD that may specifically damage the areas under investigation, especially with reference to bvFTD;
- Lines 301-311: In the way this period is formulated, it is not clear to me how the Authors intend to relate the changes in the FBI to the hippocampus in addition to the frontal region. The frontal lobes have extensive connections with many other regions, so how can one attribute frontal changes with certainty to changes in the hippocampus rather than other areas? I suggest clarifying or rewording this paragraph.
- It would be interesting, if possible, in the Discussion, to add some literature data on volumetric changes in the investigated regions and other types of FTD (i.e. right lobe variant, primary progressive aphasias), to assess any differences of interest.
Reviewer 2 Report
Comments and Suggestions for Authors
Dear Authors,
I read your work entitled “Abnormalities of hippocampal subfield and amygdalar nuclei volumes and clinical correlates in behavioral variant frontotemporal dementia with obsessive-compulsive behaviors” and here I enclose my recommendations to you:
1. There is a need for editing some of English language errors. Please, have a more thorough “look” in the text by a native speaker of English or an editing office.
2. The “Introduction” is good but there is space for improvement in order to augment the rational of this study.
3. The “Methods” section is written in a good manner, but sure extra information’s are needed. I suggest the Authors to add more info about the participants if all scales (MMSE, NPI etch.) are standardised in their language and population If not, the authors must provide rational why these scales were selected even there not standardized. Additionally the Authors must provide cut-off scores (https://pubmed.ncbi.nlm.nih.gov/34925100/) they have used to categorize their sample. Furthermore, the sample is small, and the Authors must run normality tests (Kolmogorov-Smirnoff and Shapiro-Wilk test) for the distribution of their data. The Authors must present their non-normal distributed data in Medians and IQR and non-parametric analysis must be computed (e.g., Mann-Whitney test, etch). Moreover, a sample size estimation (size effect) must be computed for their analysis (e.g., Cohen d coefficient or partial eta coefficient tech).
4. The “Results” are readers friendly and sure there is a need for new analysis (see the above comment). In Table 1 the Authors must provide which what statistical methods the proportions data and the numeric data were compared (e.g., chi-square, t-test, or Mann-Whitney test, Wilcoxon test etch.)
5. The discussion is rich, and I congratulate the Authors for that, but they must address my comments 3 and 4 in order to be more accurate. Additionally, it will be good to have a more rounded writing in this section because their sample is too small. It is also good that the Authors declared the limitations of this study.
Thank you.
Comments on the Quality of English LanguageMinor editing of English language required.
Round 2
Reviewer 1 Report
Comments and Suggestions for Authors
The Authors correctly addressed the suggestions (please correct 'dementia' in line 124).
Reviewer 2 Report
Comments and Suggestions for Authors
Dear Authors,
I read again your work entitled “Abnormalities of hippocampal subfield and amygdalar nuclei volumes and clinical correlates in behavioral variant frontotemporal dementia with obsessive-compulsive behaviors”. I congratulate you for addressing all my comments.
Please have a final check for English and some remaining syntax errors in the new texts that you have added.
Thank you.
Comments on the Quality of English LanguageMinor editing of English language required